# Exploring the use of the Psychological Characteristics of Developing Excellence (PCDEs) in younger age groups: First steps in the validation process of the PCDE Questionnaire for Children (PCDEQ-C)

**Felien Laureys** [1]*, **Dave Collins** [2], **Frederik J. A. Deconinck**[1], **Matthieu Lenoir**[1]

**1** Department of Movement and Sports Sciences, Ghent University, Ghent, Belgium, **2** Moray House School of Education and Sport, University of Edinburgh, Edinburgh, United Kingdom

* Felien.Laureys@UGent.be

## Abstract

Athletes who want to benefit most optimally and efficient from the Talent Development (TD) pathway, should start developing their psychological characteristics at a young age. The Psychological Characteristics Developing Excellence Questionnaire–Version 2 (PCDEQ2) can provide a full assessment of the mental characteristics athletes need. However, the PCDEQ2 has only been validated in adolescent athletes and as a consequence its does not contribute to the understanding of these psychological skills in a younger age group. The main purpose of this study was to take a first step in examining the factor structure and reliability of the Psychological Characteristics of Developing Excellence Questionnaire for Children (PCDEQ-C), a questionnaire based on the PCDEQ– 2. Firstly, the original questionnaire was translated to Dutch, age-adapted and redesigned for implementation in young athletes. Secondly, 774 participants (400 girls) from Flanders, Belgium, between 7 and 13 years (mean age of 10.61 ± 1.58) old filled out the questionnaire. After exploratory factor analysis, a new factor structure for the PCDEQ-C deemed an acceptable fit with 51 items in 5 factors. In the third stage, the reliability showed a good overall and internal consistency, with adequate relations between factors. The first steps in the validation process of the PCDEQ-C, suggest that this questionnaire could be a useful and reliable tool to assess the developmental psychological characteristics of 7-to-13-year-old athletes. The questionnaire is one of the first formative assessment tools to monitor and develop the psychological characteristics needed during the earliest talent development stages of a young athlete.

## Introduction

The pathway to the highest level of competition in athletic performance is dynamic, complex and challenging. Athletes not only need to overcome multiple challenges, but also have to learn the inherent lessons in order to reach the top level. This is explicitly acknowledged as the

**Data Availability Statement:** All excel files from the PCDEQ-C database will be available after acceptance on OSF.

**Funding:** FL received a grant from the Research Foundation – Flanders (FWO) with grant number FWO_3F0_2018_0031_01. The funders had no role in study design, data collection and analysis, decision to publish or preparation of the manuscript.

**Competing interests:** The authors have declared that no competing interests exist.

talent development (TD) process and the idea is supported by a wide range of literature [1]. Next to the natural abilities that future experts may possess and the environmental influences that can play a role in the TD pathway, Gagné [2] also describes the intrapersonal traits and characteristics athletes need in his Differentiated Model of Giftedness and Talent. In similar fashion, champion adult athletes indicate that having a good set of psychological skills, help them overcome and benefit from the challenges, rather than letting the challenge overtake them [3]. Importantly, however, if coaches and practitioners are committed to equipping their athletes with the best possible psychological skill set, we need more knowledge on how psychological characteristics are manifested in childhood and pre-adolescence. Without such an early basis, or data which could inform and direct remedial action, many may drop out or be blocked from progression in their sport.

Researchers have repeatedly recognised psychology as a key determinant in TD [4–6]. In fact, research has suggested for a while that psychology is an even more important factor in TD than in eventual adult performance [7]. As demonstrated by many researchers, a good psychological skillset should contain positive or adaptive characteristics such as goal-setting, self-organisation, commitment, imagery-use, self-awareness and many others [8]. With information from literature, qualitative data collection and expert opinions MacNamara and colleagues [9, 10] eventually arrived at a list of ten skills with adaptive behaviours called the Psychological Characteristics of Developing Excellence or PCDEs. The development and deployment of PCDEs should help athletes to benefit optimally from developmental challenges [11]. Next to the adaptive skills, athletes also need to be able to manage dual effect and maladaptive psycho-behavioural characteristics that influence their talent development pathway [12, 13]. The dual effect constructs can be both adaptive and maladaptive, depending on how, how much, when and in what contexts they are applied. One example of such a psychological behaviour is perfectionism, where aiming for high standards can both drive performance and/or induce negative behaviours such as eating disorders or burnout [14]. Maladaptive psychological characteristics, for example a poor mental health or having clinical issues, have a detrimental effect on the talent development process.

In light of assessing and thus, helping to develop/deploy or mitigate/eliminate these adaptive, dual effect and maladaptive psycho-behavioural characteristics, a multidimensional questionnaire for adolescent athletes was designed. The Psychological Characteristics of Developing Excellence Questionnaire–Version 2 (PCDEQ2) [1] generates seven factors, each relating to possession and application of the ten PCDEs together with other maladaptive and/or negative characteristics. The instrument consists of 88 items (Table 1). As can be seen, some PCDEs load onto multiple factors. Since the items are all gathered from literature and extensive qualitative investigation, and thus linked into a questionnaire from a theoretical point of view, this seems inevitable. This is especially so since they are usually administered in small subgroups, offering the athlete with a bespoke solution to each specific challenge [15]. Reflecting this, the questionnaire is used as a formative tool to help athletes and coaches figure out relative strengths and weaknesses which can then be addressed.

Importantly, however, the PCDEQ2 has only been validated with adolescent athletes. Consequently, the PCDEQ2 does not contribute to our knowledge of the understanding and application of these psychological skills in other contexts, such as younger age groups. Nevertheless, if athletes want to benefit from the TD pathway in the most optimal and efficient way, coaches should start developing the psychological characteristics of their athletes at a young age. Indeed, in early specialization sports (such as gymnastics or swimming), where athletes already have a high training volume before puberty and an early age of peak performance [16], psychological skills are already of crucial importance at a pre-adolescence age. Previous research has already examined to what extent young athletes between 10 and 15 years old implicitly

**Table 1. Factors and their underlying PCDEs of the PCDEQ2 (Hill et al., 2019 [1]).**

| Factors | | PCDEs |
|---|---|---|
| 1 | Adverse response to failure | Fear of failure |
| | | Focus and distraction control |
| | | Perfectionism |
| | | Anxiety-related behaviours |
| 2 | Imagery and active preparation | Realistic and controllable imagery |
| 3 | Self-directed control and management | Planning and organisation |
| | | Self-regulation and self-control |
| 4 | Perfectionistic tendencies | Perfectionism |
| | | Passion |
| | | Fear of failure |
| | | Anxiety-related behaviours |
| 5 | Seeking and using social support | Creating and using support networks |
| 6 | Active coping | Coping with pressure |
| | | Self-regulation and self-control |
| 7 | Clinical indicators | Anxiety-related behaviours |
| | | Depressive symptoms |
| | | Eating disorders |
| | | Behavioural change |

PCDEs: Psychological Characteristics of Developing Excellence.

understood the constructs of psychological skills [17]. These studies suggested that the athletes in general had some understanding of these skills, with older athletes clearly having a better knowledge of the psychological constructs than the younger ones. Furthermore, research has already shown that, although young children might not comprehend the terms used in questionnaires or by experts, this does not mean they are not using the underlying psychological characteristics already in their athletic career. An example for this is imagery. Imagery is a difficult term to understand without explanation, but the skill is frequently used, although perhaps with differences in vividness and underlying cognitive abilities, by young athletes [18–20]. This reflects a possible developmental difference in the knowledge, understanding and use of the psychological skills which would benefit from empirical examination and, if appropriate, remediation.

Next to the comprehension issue in younger athletes, the PCDEQ2 should not be copied and used in younger, preadolescent athletes without examining both the comprehension and factor structure in this age group. According to previous literature [21], development is non-linear and assessment of any kind or form of expertise should therefore be adjusted to the developmental phase the athlete is currently in. For psychological skills, athletes will be exposed to different needs in the initiation, development and mastery stage [5]. As athletes continue their developmental process, the demands of the environment become more difficult or change in type and balance. Consequently, although the fundamentals of the PCDEs could be present in the preadolescent group, not all PCDEs may be as valuable or meaningful as in the older age group, therefore adding to the potential for a different factor structure in the preadolescent athletes.

Therefore, the main purpose of this study was to adapt the PCDEQ2 to a new Dutch questionnaire based on the PCDEQ2 but for children between 7 and 13 years old, called the Psychological Characteristics of Developing Excellence Questionnaire for Children (PCDEQ-C).

To achieve this eventual purpose, some first exploratory steps towards the validation of the PCDEQ-C were taken in this study. First, to ensure that the young athletes successfully understood the items as they were originally intended [22], cognitive interviewing was applied with respect to comprehensibility of the items, with adjustments being made where appropriate. Second, the factor structure of the PCDEQ-2 [1] was studied by means of an exploratory factor analysis. Finally, the reliability of the PCDEQ-C was examined, using a test-retest method with the preadolescent athletes. For clarity, the method and results are presented together for each of these distinct and sequential stages.

## Method and results

### Stage 1: Item generation and confirmation

**Method.** *Item generation*. As the first step, the original PCDEQ2 was translated and adapted into a Dutch version. For this translation process, a collaboration between the researchers from Ghent University, Belgium and the original authors of the PCDEQ2 was established. Accordingly, native Dutch and English speakers were included in the translation process. The original PCDEQ2 focused on 14-to-19-year-old male athletes in team sports and needed to be adapted to 7-to-13-year-old male and female athletes. A list of the factors and the corresponding items can be found in the article of Hill et al. [1]. To make the PCDEQ-C useful for a broad range of sports, both team and individual, and generic items were included.

First, the appropriateness of the English items in the original questionnaire (the PCDEQ2) was checked by the Belgian-British panel. Items that were too difficult for 8-year-olds were rephrased in English. This only led to rewriting 3 items, specifically because of the term 'visualisation'. Next, the English items were translated into Dutch. The translation was kept as close to the initial phrasing as possible, although some sentences had to change because of grammatical reasons. One item was removed, because this appeared twice in the questionnaire. This resulted in a first Dutch version, called the PCDEQ-C with 87 items.

*Cognitive interviewing*. Cognitive interviewing was conducted applying the principles developed by Willis and colleagues [22], using think-aloud protocols, reinterpretations, observations with proactive and reactive verbal probing. Individual interviews were conducted with three 8-year-old female gymnasts while they completed the PCDEQ-C. These gymnasts already trained up to 6 hours per week and were deemed experienced in practicing their sport.

Since this is a rather elaborate questionnaire (87 items), a short break after each page was allowed for the participants (more or less after 10 items). Participants were asked to read the items out loud, then a combination of protocols were used by the researchers to make sure that the participant fully understood the item. Questions were asked relating to explaining the meaning of certain words, giving examples of what the item meant or asking what score they would give themselves. Answers to those questions were noted, together with observations of the behaviour of the participants while filling-in the questionnaire (hesitations, confusion expressions, skipping an answer).

*Back-translation*. Once these adaptations were made, a back-translation was made from Dutch to English, by a native Dutch researcher not involved in the study. This translation was then compared with the original English questionnaires by the native Dutch and English researchers. Only three small comments were made, all of which were adopted.

**Results.** Based on the cognitive interviews and back-translation, 28 items were changed, because the item itself or the words used in the items were too difficult for young children to understand (e.g. self-discipline). Sometimes only the sentence structure was changed, other times another word or an explanation of the original word was used. These adaptations apart, however, the questionnaire then deemed appropriate for further analysis.

## Stage 2: Exploratory factor analysis

**Method.** *Participants*. The participants in this study were young sports athletes between 7 and 13 years old (mean age of 10.61 ± 1.58) from Flanders, the Dutch-speaking part of Belgium. The athletes were contacted through elementary and high schools or through sports clubs and were asked to fill in the questionnaire. They reported if they practiced a particular organised sport, and if so, which sport and how many practice hours per week per sport. This resulted in 1200 filled-out questionnaires. However, only individuals who fully completed the questionnaire (a clear answer on each item) and who practiced an organised sport for least 3 hours per week at a recreational or competitive level, were included in this study. This resulted in a total sample of 773 participants (400 girls), with 495 individual athletes (e.g. gymnastics, cycling and badminton) and 278 team sports (e.g. soccer and hockey) athletes. A detailed overview of the distribution per age, sex, type of sport and level can be found in S1 Table. All participants gave informed consent via their parents or legal guardian, as they were under the age of 18. The Ethics Committee of Ghent University Hospital approved this study.

*Procedure*. Participants were asked to fill out a paper version of the questionnaire while thinking about their main sport and their experiences during the past few weeks. The questionnaire consisted of 87 items on a 6-point Likert scale, ranging from 1 ("very unlike me") to 6 ("very like me"). Notice that this is a self-report and self-perception type of questionnaire. Participants had to work individually; however, they were in the same room with their team or class. Here as well, participants completed the questionnaire in their own time and were allowed to take short breaks in between. A test leader was always present to answer possible questions. For the younger ages, a test leader read the questions out loud. A pilot study had indicated that the reading skills of 8-year-old-children were not always sufficient to answer each item independently. By reading the items out loud, it was easier for the younger participants to understand the items and to give an accurate answer. It took the participants about 20 to 40 minutes to fill in the questionnaire, depending on their age. To avoid impression management (i.e. some participants purposefully fabricate an answer to create the most positive social image which would impress his/her audience) [1], and in accordance with other conversional psychological and/or emotional assessments with children [23], no parents or coaches were present at the moment of filling-in the questionnaire.

*Data analysis*. Before performing the factor analysis, we first checked if the sample size was adequate. Therefore, Kaiser-Meyer-Olkin (KMO) measurements and Bartlett's test of sphericity were used. The sample size is regarded appropriate if the KMO value is as close to 1 as possible, or at least above 0.5 [24], and if the p-value for the Bartlett's test of sphericity is at least significant to the 0.05 level [25].

To examine the factor structure, first the item-to-total correlation was checked, as recommended by Field [26]. Items with a correlation below 0.3 were left out of the further analysis [27]. Next, a parallel analysis (PA) was performed to suggest the number of factors that would give the best fit. The parallel analysis is considered the "golden standard" to use when determining the numbers of factors [26]. Based on the PA, EFA, with an oblique rotation and the minimal residual method, was used with the suggested number of factors but also, offering a second level check, with more or less factors. This approach was followed to end up with the best possible fitting model, since PA often gives an over- or underestimation of the number of factors [28]. The model fit measures used for the EFA, were the chi square ($\chi^2$) and degrees of freedom (df), $\chi^2$/df with the p-value, comparative fit index (CFI), the Tucker-Lewis index (TLI), root mean square error of approximation (RMSEA) and the standardised root mean square residual (SRMR). To have a good model fit, the following criteria were used: $\chi^2$/df $<3$ [28], CFI and TLI $>0.90$ [29], RMSEA and SRMR $<0.05$ [30]. All analyses were performed in

**Table 2. Goodness of fit indices for each model (M) of the exploratory factor analyses (EFA) with a 5, 6, 7, 8 and 9 factor structure (N = 774).**

| Model | $\chi^2$ (df) | $\chi^2$/df | p | TLI | RMSEA | SRMR |
|---|---|---|---|---|---|---|
| M1<br>5 factors | 3088.0 (1765) | 1.75 | <0.001 | 0.874 | 0.031 | 0.030 |
| M2<br>6 factors | 3329.3 (1885) | 1.77 | <0.001 | 0.882 | 0.031 | 0.030 |
| M1<br>7 factors | 3087.6 (1823) | 1.70 | <0.001 | 0.893 | 0.029 | 0.030 |
| M2<br>8 factors | 2860.7 (1762) | 1.53 | <0.001 | 0.904 | 0.028 | 0.030 |
| M3<br>9 factors | 2645.4 (1702) | 1.55 | <0.001 | 0.914 | 0.026 | 0.020 |

$\chi^2$ = Model Chi Square; df = degrees of freedom; TLI = Tucker-Lewis Index; RMSEA = Root Mean Square Error of Approximation; SRMR = Standardized Root Mean Square Residual.

R (version 3.4.4; Team, 2013); with the Lavaan and Psych R Packages (version 0.5–12 and version 1.9.12; respectively) [31, 32].

**Results.** According to the KMO (0.9) and Bartlett's test of sphericity (P < 0.01), the sample size was sufficient and therefore appropriate to perform further analyses. In total, 22 items had an item-to-total correlation lower than 0.3. The PA was thus performed with 65 items. The PA suggested that for the 87-items questionnaire, 8 factors should be used. Accounting for the possible under- and overrepresentation of the PA, the EFA was used with 5, 6, 7, 8 and 9 factors. The different structures were examined and compared, taking into account the following criteria: only items with loadings above 0.3; no or few cross-loadings items and no factors with less than three items [26]. Although fit measures for the 5 factor structure were the weakest (Table 2), this 5-factor model was seen as the best fit, as this had almost no cross-loadings and was the only model providing at least three items per factor. From the 65 items originally in the EFA model, 12 items with an item loading under 0.3 were removed [26]. This model ended with 53 items corresponding to 5 factors (Table 3 and S2 Table).

Next, the interpretation of each item in the 5-factor solution was further inspected. From a statistical point of view, items with cross-loadings or negative loadings were closely inspected, to determine their fit into the intended factor. The model was also checked on its theoretical basis, i.e., are the items, with its underlying PCDE's, structured in a theoretically logical manner. This resulted in switching 9 items to another factor and removing 2 items because of a better theoretical fit. In total, the 5-factor solution ended with 51 items. Although the TLI and CFI of the 5-factor model indicates a statistical weak model, other fit measures were acceptable and this new model is now also acceptable in terms of its meaningfulness, as it is in line with other work in this field [1, 7].

Experts also examined the factor structure with the remaining 51 items, with the aim of proposing factor titles. Eventually, 5 factor names were chosen, based upon the highest item loadings (above 0.4) within each factor. In naming each factor, we followed the usual convention of focusing on the content [28, 33]. As a secondary consideration, however, we tried to stay as close as possible to the factor names selected for the PCDEQ2 [1] (S3 Table).

Factor 1, *Performance Worries (N = 16)*, incorporates items related to (performance) anxiety. Athlete's with high scores on this factor are not only risking deteriorating their talent development pathway but also their own well-being [11, 34]. High levels of anxiety could potentially lead to avoidance-based coping behaviours, in turn activating a vicious circle of

**Table 3. Factor loadings for 53-item Psychological Characteristics of Developing Excellence Questionnaire for Children.**

| Item | Factor 1 | Factor 2 | Factor 3 | Factor 4 | Factor 5 |
|---|---|---|---|---|---|
| Q28 | 0.722 | | | | |
| Q46 | 0.628 | | | | |
| Q37 | 0.599 | | | | |
| Q72 | 0.583 | | | | |
| Q21 | 0.567 | | | | |
| Q49 | 0.538 | | | | |
| Q42 | 0.536 | | | | |
| Q27 | 0.531 | | | | |
| Q22 | 0.525 | | | | |
| Q52 | 0.521 | | | | |
| Q17 | 0.520 | | | | |
| Q3 | 0.477 | | | | |
| Q40 | 0.477 | | | | |
| Q86 | 0.476 | | | | |
| Q75 | 0.459 | | | | |
| Q38 | 0.450 | | | | |
| Q7 | 0.448 | | | | |
| Q34 | 0.446 | | | | |
| Q48 | 0.412 | | | | |
| Q83 | 0.374 | | | | |
| Q45 | 0.340 | | | | |
| Q13 | 0.327 | | | | |
| Q41 | 0.301 | | | | |
| Q71 | | 0.622 | | | |
| Q81 | | 0.467 | | | |
| Q69 | | 0.464 | | | |
| Q59 | | 0.439 | | | |
| Q73 | | 0.374 | | | |
| Q79 | | 0.364 | | | |
| Q51 | | 0.340 | | | |
| Q64 | | | 0.656 | | |
| Q44 | | | 0.651 | | |
| Q68 | | | 0.645 | | |
| Q33 | | | 0.615 | | |
| Q15 | | | 0.570 | | |
| Q32 | | | 0.562 | | |
| Q78 | | | 0.555 | | |
| Q63 | | | 0.505 | | |
| Q58 | | | 0.501 | | |
| Q25 | | | 0.492 | | |
| Q9 | | | 0.449 | | |
| Q23 | | | 0.442 | | |
| Q24 | | | 0.421 | | |
| Q30 | | | 0.367 | | |
| Q60 | 0.323 | | | 0.415 | |
| Q36 | | | | 0.389 | |
| Q39 | | | | 0.377 | 0.334 |

(*Continued*)

**Table 3.** (Continued)

| Item | Factor 1 | Factor 2 | Factor 3 | Factor 4 | Factor 5 |
|------|----------|----------|----------|----------|----------|
| Q82 |  |  |  | 0.337 |  |
| Q87 |  |  |  |  | 0.503 |
| Q85 |  |  |  |  | 0.470 |
| Q47 |  |  |  |  | 0.459 |
| Q54 |  |  |  |  | 0.398 |
| Q4 |  |  |  |  | 0.363 |

increased anxiety and avoidance [35]. Factor 2, *Social support (N = 9)*, is related to items concerning using and seeking social support of others. Athletes benefit from an environment with a positive supporting network, helping and even facilitating them with the inherent challenges in the TD pathway [1, 36]. Factor 3, *Imagery and Active Preparation (N = 13)*, has high-loading items on imagery use during practice and performance. It also contains components of goal setting behaviour, required in TD [1, 7]. Factor 4, *Adverse Response to Failure (N = 7)*, primarily highlights the maladaptive response to failure, especially with regards to negative social evaluations [35, 37]. This factor also concentrates on items related to eating disorders. This comes as no surprise, given the relation that is found between fear of failure and eating disorders [38]. Factor 5, *Self-Directed Control and Management (N = 6)*, plays an opposite role in the PCDE's, concerning the positive use of planning, organisation, self-control and self-regulation [1].

## Stage 3: Reliability

**Method.**   *Participants*. For the reliability check, a small sample of the participants from Study 2 were used. Gymnasts and football players were contacted through convenience sampling in different clubs, to participate in this test-retest design. In the gymnastics population, only the recreational gymnasts were asked to fill in the questionnaire twice, because of arrangements with the clubs and Gymnastics Federation. The recreational gymnasts were between 7 and 13 years old. From the 55 recreational gymnasts participating in the test-retest procedure, 43 gymnasts completed the questionnaire twice. In the football clubs, the U8 to U12 football teams were tested and 83 participants completed the test-retest procedure. In total 126 athletes were included in the test-retest procedure. All participants gave informed consent via their parents or legal guardian, as they were under the age of 18. The Ethics Committee of Ghent University Hospital approved this study.

*Procedure*. To check the reliability of the PCDEQ-C after the factor structure was established, a test-retest procedure was set-up. The interval between the test-retest moments should not be too small, because the participants could still remember their answers, but it should also not be too large, so that the conditions were still comparable [39] and athletes' perception would not change [40]. Accordingly, we decided that participants completed the questionnaire twice, with a two-week rest period in between. When the assessment took place at the end of a training session, participants were asked to stop the training session half an hour earlier. Participants came in half an hour before the start of their session, when the assessment was before their training session. At both times, the participants filled out the questionnaire in an available space at the club site (dressing or meeting room, depending on the club). Once again, a researcher read the questions out loud for the younger ones at both test moments and only a researcher was present with no coaches, parents or other caregivers.

**Table 4. Mean (M) and standard deviation (SD) of the two test moments per factor, and the difference in mean between time point 1 and 2 (ΔM), also the McDonald Omega (ω) and Pearson correlation (r) between the items per factor.**

| | Number of items | Time 1 | Time 2 | ΔM | t | Ω | r |
|---|---|---|---|---|---|---|---|
| | | M (SD) | M (SD) | | | | |
| **Factor 1** *Performance Worries* | 16 | 3.39 (0.88) | 3.17 (0.88) | 0.22 | 4.072* | 0.86 | 0.76 |
| **Factor 2** *Social support* | 9 | 4.37 (0.84) | 4.33 (0.87) | 0.04 | 0.619 | 0.73 | 0.68 |
| **Factor 3** *Imagery and Active Preparation* | 13 | 4.10 (0.95) | 3.98 (1.02) | 0.12 | 1.979* | 0.84 | 0.76 |
| **Factor 4** *Adverse Response to Failure* | 7 | 2.86 (1.11) | 2.69 (0.97) | 0.17 | 2.414* | 0.75 | 0.70 |
| **Factor 5** *Self-Directed Control and Management* | 6 | 4.21 (0.86) | 4.32 (0.90) | - 0.11 | - 1.526 | 0.66 | 0.60 |

* P < 0.05; N = 126.

*Data analysis*. McDonald omega value was used as a measure for internal consistency within and between the factors. Omega values can range between 0 and 1, where a value closer to 1 and above 0.70 is considered to represent a good reliability [41, 42]. Afterwards, t-test, Omega and correlations between the factors of test moment 1 and 2 were examined to explore potential differences between the test-retest moments of the recreational gymnasts and football players.

**Results.** The overall internal consistency is considered good with an Omega of 0.80. The Omega per factor was also good, with reliability values between 0.66 and 0.86 (Table 4). The internal consistency was further supported with the data from the test-retest procedure. For factor 1, 3, and 4 good correlations between test moment 1 and 2 were found (between 0.70 and 0.76). For factors 2 and 5, correlations between 0.60 and 0.68 were seen, considered as rather weak correlations. Paired-Samples T-test also showed a significant difference between time point one and two for factor 1, 3 and 4. Overall, this finding is relatively good.

## Discussion

To increase our knowledge on the psychological characteristics in the talent development process, this study provided first steps in the validation of a children's version of the PCDEQ2, called the PCDEQ-C. After item generation exploring a new factor structure, it seems that the PCDEQ-C is a questionnaire, with 5 factors concerning 51 items. Reliability, using a test-retest procedure, also showed positive results. Based on the findings, statistical and theoretical rationale shows support for a 5-factor structure of the PCDEQ-C, a formative assessment tool that can be used with athletes between 7 and 13 years old, in their young, athletic careers.

Since psychological characteristics already play an important role in the first stages of the talent development process [5], a tool was needed to assess and monitor these skills and characteristics. Therefore, a first step was taken to develop the PCDEQ-C for athletes between 7 and 13 years old. This questionnaire is a precursor of the PCDEQ, which is usable for athletes between 14-to 19–year-olds. Firstly, an EFA examined the factor structure of 65 items. Results showed that a 5-factor structure could be the best possible solution. However, it must be said that an EFA does not present the whole truth and can sometimes have a more subjective nature [43]. Moreover, we wanted the final model to stay as close as possible to the factor structure of Hill and colleagues [1], for future follow-up in the developmental trajectory of athletes.

Although the 5-factor solution with 51 items gave a slightly lower fit than the EFA solution, it was still deemed an acceptable fit, supporting the idea that the PCDEQ-C is usable in this young, athletic population. It is important however, that the factor structure proposed by the EFA, but slightly adapted by the theoretical rationale, should be further examined and confirmed in younger age group contexts.

The 5 factors proposed here, suggests that coherence can change over age, going from a 5-factor structure in childhood athletes to a 7-factor structure in adolescent athletes [1]. The developmental nature of the PCDEs is further supported when looking at the low values for the means of the five factors. Since athletes between 7 and 13 years old are only at the start of their athletic development, low scores on the factors do not need to be alarming or taken to imply these athletes will never make it to the top. This simply means that the PCDEs are, as the name itself implies, characteristics that need to be developed. As is also observed in this study, Blijlevens and colleagues [5] suggest that young athletes already possess psychological characteristics and skills at a relatively young age. Furthermore, several researchers [5, 7] suggest that athletes who have a range of psychological skills at a young age, even at a low level, will be able to more effectively use this in later stages of their talent development process to overcome certain obstacles. Additionally, the skill-set that athletes need, will only increase to efficiently and positively cope with the increasing demands on other areas in the expertise development.

The reliability in this study, evaluated with a test-retest procedure, showed a moderate to good reliability between the test and retest. Although there was a significant decline in the mean score between time point one and time point two on some factors, the difference in means is relatively small. One possible reason could be the way the Likert-scale was presented to the children (from "very like me" to "very unlike me"), which could be difficult for young children, especially with negative-worded items. Another reason could be a shift in emotional state or energy level, that could possibly mediate/moderate the difference in answers. However, the means of both time points on all the factors are very close and have no difference at a practical level. Either way, caution is needed when assessing the PCDEQ-C, especially with under 10-year-olds. During testing, it was observed that filling-out the questionnaire was difficult for the youngest participants. Although they understood the items in the questionnaire, they were more comfortable providing answers when a researcher read the items out loud. We advise to complete the assessment of the PCDEQ-C in a quiet room, with a one-on-one guidance, to make the responses of the young athletes as accurate as possible.

To further validate the PCDEQ-C, more information about the understanding and application of PCDEs in different contexts is necessary. Planning and organisation, one of the PCDEs, is already of importance during childhood. However, younger athletes will likely rely on their parents and coaches for their planning, without giving it further thought. At a later stage, when athletes are older and further down their talent development pathway, they will make the planning in co-creation with the coach [44]. Next to age, differences in sex in these younger age groups needs further examination. There are for example sex-based differences in the prevalence of certain psychopathological problems in adolescent athletes, such as eating or anxiety disorders, two maladaptive characteristics also used in the current study [45]. Sex differences may also covary with maturation, as girls at this stage may often be ahead of boys both physically and emotionally [46]. Examining the differences between young boys and girls on not only these maladaptive, but also positive psychological characteristics, is thus necessary. Coinciding with age- and sex differences, therefor another factor that needs recognition, is the type of sports young athletes participate in. Next to the typical differences in team sports vs individual sports, already seen in the adolescent age group [6], athletes from the same age could be in different talent developmental stages, depending on their sport. Early specialisation sports (e.g. swimming, gymnastics,.) require a higher and more complex set of (psychological)

skills at younger ages than normal or late specialisation sports (e.g. badminton, soccer, . . .). At the moment, only age is considered in the development of the PCDEs, although perhaps more information will come when approaching the PCDEs according to the talent development stage.

Some limitations regarding the validation process of the PCDEQ-C should be addressed. First, the PCDEQ-C is now based on age-cut offs, although it could be interesting to investigate if cognitive maturity level could be a better indicator for understanding and using the PCDEs. It should also be recognised that the questionnaires regarding the PCDEs are filled-in by the athletes themselves. Next to the general issues with self-report, special care should be given to the competitive nature of talent development environments, where individuals potentially employ impression management strategies when responding to any questionnaire. Lastly, although overall reliability is considered adequate, low reliability scores were observed on two factors. When follow-up is provided on the results of the PCDEQ-C, special attention should be given to these two factors.

Since this was only a first step in the validation process, further research is necessary. Further work should investigate the confirmation of the proposed factor structure, in the different contexts as suggested, and with large sample sizes. More types of sport should be included, so future work could also investigate the discriminating ability in type of sport. Other information regarding training history, training intensity, competitive achievements and/or other sociodemographic variables should also be explored. Next the application of the PCDEs in these younger ages also needs further attention, especially against self-presentational bias. An example thereof is to investigate to what extent the answers the young children gave are in line with what parents or coaches observe; in short, to evaluate and confirm the self-reflection capacity of young children. Finally, this study is also cross-sectional in nature. Longitudinal study designs are necessary to follow-up of the development of the PCDE's and gain knowledge of the psychological skills and characteristics that contribute to specific stages of the talent development process.

The present study provides a first step in the validation process of a formative assessment tool for the developmental psychological characteristics in athletes between 7 and 13 years old, called the PCDEQ-C. Rather than a talent selection tool, this questionnaire should be used as a formative assessment tool to monitor, develop, and design the psychological characteristics needed during the entire TDE pathway of a young, potential athlete. The questionnaire is being increasingly reported as a practical and comprehensive tool to measure PCDEs by both athletes and coaches [47] and should be used as part of the triangulation process. The use of other measures, such as behavioural observations, coaches' opinions, dialogue with family or guardians, and follow-up of the PCDEQ-C results by coaches and/or sport psychologist are advised. More work needs to be done to further validate the PCDEQ-C, so the PCDEs can be used during the entire athletic career. In the early stages of the athlete before puberty, the athletes would then be able to fill in the PCDEQ-C, with a limited number of items and factors that will suit the age range of 7 to 13 years old best. As the athlete grows older, the set of psycho-social skills and behaviours will increase in number and difficulty, as the demand of the environment will increase in difficulty as well. During these years (14 and onwards), the PCDEQ2 can used. Both are based on the same set of psychological characteristics (PCDEs), having an adaptive effect, maladaptive effect or dual-effect on the athletic development.

## Supporting information

**S1 Table. Distribution per age and sex, type of sport and level.**
(PDF)

**S2 Table. Items per factor of the PCDEQ-C.**
(PDF)

**S3 Table. Comparison of the PCDE factors for the PCDEQ-C and PCDEQ-2.**
(PDF)

## Author Contributions

**Conceptualization:** Felien Laureys, Dave Collins, Frederik J. A. Deconinck, Matthieu Lenoir.

**Formal analysis:** Felien Laureys.

**Funding acquisition:** Dave Collins, Matthieu Lenoir.

**Investigation:** Felien Laureys.

**Methodology:** Felien Laureys, Dave Collins.

**Project administration:** Dave Collins, Matthieu Lenoir.

**Software:** Felien Laureys.

**Supervision:** Dave Collins, Frederik J. A. Deconinck, Matthieu Lenoir.

**Validation:** Dave Collins.

**Visualization:** Frederik J. A. Deconinck.

**Writing – original draft:** Felien Laureys.

**Writing – review & editing:** Dave Collins, Frederik J. A. Deconinck, Matthieu Lenoir.

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
