## [Decision Letter · Decision Letter 0]

2 Sep 2021

PONE-D-21-19108

Exploring the use of the Psychological Characteristics of Developing Excellence (PCDEs) in younger age groups: first steps in the validation process of the PCDE Questionnaire for Children (PCDEQ-C)

PLOS ONE

Dear Dr. Laureys,

Thank you for submitting your manuscript to PLOS ONE. After careful consideration, we feel that it has merit but does not fully meet PLOS ONE’s publication criteria as it currently stands. Therefore, we invite you to submit a revised version of the manuscript that addresses the points raised during the review process.

We look forward to receiving your revised manuscript.

Kind regards,

Frantisek Sudzina

Academic Editor

PLOS ONE

Journal Requirements:

Reviewers' comments:

Reviewer's Responses to Questions

**Comments to the Author**

1. Is the manuscript technically sound, and do the data support the conclusions?

Reviewer #1: Yes

2. Has the statistical analysis been performed appropriately and rigorously? 

Reviewer #1: Yes

3. Have the authors made all data underlying the findings in their manuscript fully available?

Reviewer #1: Yes

4. Is the manuscript presented in an intelligible fashion and written in standard English?

Reviewer #1: Yes

5. Review Comments to the Author

Reviewer #1: Dear authors,

The work carried out represents an important contribution to the field of knowledge. However, in the opinion of the reviewer, a series of modifications is necessary for the paper to be published.

Thanks for your effort

The research carried out is adequate and well justified.

The keywords are well selected.

Line 38.- The year is missing in Gagné

Line 47 and 48.- “Researchers have repeatedly recognized psychology as a key determinant in TD 48 (4-6)” It is recommended to include references from MacNamara and Collins 2011 (PCDEQ).

Reflection: Can talent be detected with 7-year-old athletes? Are these children ready for a self-report assessment?

Such a low age range is not adequately justified for self-report psychological tests. Most of the tests used in Sports Psychology are for athletes aged 12 years or older due to: intellectual abilities developed close to adulthood; development of self-concept (among others).

Specify that the PCDEQ questionnaire is a self-perception and self-report questionnaire, not based on an objective analysis of the reality that surrounds the athlete (evaluation by experts, such as coaches; evaluation of the family, parents or guardians; not based on aptitude tests) .

Does the test have the capacity to detect inappropriate or maladaptive behaviors such as perfectionism or burnout?

Is the adaptation created for both individual and collective sports valid?

Isn't it too long a test with 87 items for validation?

164.- “Results Based on the cognitive interviews and back-translation”. To include references.

171.- Method

The characteristics of the participants are not sufficiently specified. Indicate the type of sport practiced, the weekly practice time, the sporting experience, if they are competitors or not, how many are federated

The distribution of the sample by age and sex is not shown.

It is important to specify that the questionnaire refers to your experiences a few weeks ago, and not considering the total time as an athlete.

The statistics used are correct.

It is not understood why an exploratory factorial study has not been carried out to observe the resulting number of factors. The criterion of factors with eigenvalues greater than 1 and the analysis of the slope of the factors has not been included.

The test with such a different number of factors is not well understood.

It is not understood why the reduction of the items was not carried out before the administration of the test, and they are eliminated afterwards.

The factors extracted are very far from the theoretical model initially proposed by the PCDEQ questionnaire (it does not include factors related to anxiety, but it does include factors related to coping).

The Imaginary factor is not well understood in 7-year-olds. Self-assessment of visualization ability in a 7-year-old and a 13-year-old cannot be equipped.

What level of control and clarity does the 7-year-old have to visualize a game or training action?

It is not clear how many compete, at what level they compete, and what performance is achieved.

The administration of questionnaires to gymnastics and soccer participants is not representative of the rest of the sports (there are not combat sports; there are not racket sports; there are not sports such as athletics).

Two weeks in the test-retest is a very short period of time. The adapted questionnaire is a questionnaire based on a perception of the situational athlete. This data makes it difficult to establish long-term performance predictions. 

Why is the Omega reliability index used and not the Cronbach's Alpha coefficient?

Test-rest correlations obtained below .70 are not adequate. They are small, and should be considered a limitation in the study. This data indicates the low temporal stability of the scores obtained, limiting the ability to establish medium and long-term predictions. With test-restest administrations lasting approximately one month, it is very likely that the correlations were much lower.

Further development of the limitations of the study is lacking.

Further development of the practical implications of the study for sports coaches and psychologists is lacking.

Further development of future researchlines to follow is lacking.

Important bibliographic references are missing in the paper.

Differences in scores have not been established based on age, sport age category, performance level and category by sex. The discriminatory ability of the test cannot be determined.

It is not understood why the test has been administered to athletes aged less than 11-12 years. There are basic theoretical and methodological factors that indicate the inappropriateness of administering self-report questionnaires with ages less than 12 years.

It is not understood why psychopathological factors are introduced when the test does not measure psychopathological constructs or other psychopathological questionnaires have been administered.

It is noteworthy how the authors, knowing the importance of the evolution of talent and not talent at a specific moment, have not carried out a longitudinal study (with fewer participants and with a relatively short time).

Why is the factorial structure not maintained with respect to the previous tests?

The implications of the test for a sports psychologist are not included at any time, and yet psychopathological terms are included.

The reviewer considers that for the publication of the article the following should be included:

1.- The initial and final test items.

2.- With such a high number of athletes evaluated, it would be advisable to include the corresponding scales.

6. PLOS authors have the option to publish the peer review history of their article (what does this mean?). If published, this will include your full peer review and any attached files.

Reviewer #1: **Yes: **Roberto Ruiz-Barquín (Universidad Autónoma de Madrid; Spain)

---

## [Author Response · Author response to Decision Letter 0]

15 Oct 2021

Dear editor and reviewer,

We are pleased to get the opportunity to revise our manuscript. We would like to thank you and the reviewer for the insightful comments and constructive feedback. In this rebuttal (Response to the reviewers) we will respond to each comment and provide a point by point description of how each comment was addressed in the manuscript. The reviewer can follow the changes made in the manuscript, the lines described in the document below are based on the Revised Manuscript with Track Changes.

No references were removed in the revised manuscript, but two references were added:

Munroe-Chandler KJ, Hall CR, Fishburne GJ, Strachan L. Where, when, and why young athletes use imagery: An examination of developmental differences. Research Quarterly for Exercise and Sport. 2007;78(2):103-16.

Uphill MA, Lane AM, Jones MV. Emotion Regulation Questionnaire for use with athletes. Psychology of Sport and Exercise. 2012;13(6):761-70.

---

## [Editor Report · Decision Letter 1]

19 Oct 2021

Exploring the use of the Psychological Characteristics of Developing Excellence (PCDEs) in younger age groups: first steps in the validation process of the PCDE Questionnaire for Children (PCDEQ-C)

PONE-D-21-19108R1

Dear Dr. Laureys,

We’re pleased to inform you that your manuscript has been judged scientifically suitable for publication and will be formally accepted for publication once it meets all outstanding technical requirements.

Kind regards,

Frantisek Sudzina

Academic Editor

PLOS ONE
---

## [Editor Report · Acceptance letter]

5 Nov 2021

PONE-D-21-19108R1 

Exploring the use of the Psychological Characteristics of Developing Excellence (PCDEs) in younger age groups: first steps in the validation process of the PCDE Questionnaire for Children (PCDEQ-C) 

Dear Dr. Laureys:

I'm pleased to inform you that your manuscript has been deemed suitable for publication in PLOS ONE. Congratulations! Your manuscript is now with our production department. 

Kind regards, 

on behalf of

Dr. Frantisek Sudzina 

Academic Editor

PLOS ONE